# Fructose overconsumption impairs hepatic manganese homeostasis and ammonia disposal

Jian-Hui Shi[1,3], Yu-Xia Chen[1,3], Yingying Feng[1,2,3], Xiaohang Yang[2,3], Jie Lin[1], Ting Wang[2], Chun-Chun Wei [1], Xian-Hua Ma[1], Rui Yang[1], Dongmei Cao[1], Hai Zhang[1], Xiangyang Xie [2], Zhifang Xie [1] & Weiping J. Zhang [1,2] ✉

Arginase, a manganese (Mn)-dependent enzyme, is indispensable for urea generation and ammonia disposal in the liver. The potential role of fructose in Mn and ammonia metabolism is undefined. Here we demonstrate that fructose overconsumption impairs hepatic Mn homeostasis and ammonia disposal in male mice. Fructose overexposure reduces liver Mn content as well as its activity of arginase and Mn-SOD, and impairs the clearance of blood ammonia under liver dysfunction. Mechanistically, fructose activates the Mn exporter Slc30a10 gene transcription in the liver in a ChREBP-dependent manner. Hepatic overexpression of Slc30a10 can mimic the effect of fructose on liver Mn content and ammonia disposal. Hepatocyte-specific deletion of Slc30a10 or ChREBP increases liver Mn contents and arginase activity, and abolishes their responsiveness to fructose. Collectively, our data establish a role of fructose in hepatic Mn and ammonia metabolism through ChREBP/Slc30a10 pathway, and postulate fructose dietary restriction for the prevention and treatment of hyperammonemia.

Manganese (Mn) is an essential trace element for the activities of cellular enzymes, such as arginase, glutamine synthetase, pyruvate carboxylase and manganese superoxide dismutase (Mn-SOD)[1]. As a cofactor of numerous metalloproteins, Mn plays important roles in the regulation of development, digestion, reproduction, antioxidation, immune response, neuronal activity, and energy production. Because Mn is a ubiquitous metal found in a variety of food (fruit, nuts, legumes, whole grains, and brewed tea), overt Mn deficiency is generally not clinically recognized in humans. Mn deficiency has been observed in laboratory animals and has been associated with impaired growth, skeletal defects, reduced reproductive function, birth defects, and abnormal glucose tolerance, as well as altered lipid and carbohydrate metabolism[2,3]. In contrast, Mn poisoning may be encountered upon overexposure to this metal. Excessive Mn tends to accumulate in the liver, pancreas, bone, kidney, and brain, with the latter being the

major target of Mn intoxication. Deficiency or overload of Mn in the cells can cause the wide range of disorders of the physiological process, as well as the development of metabolic diseases[1,4].

Mn levels in the tissues are maintained stable via tight homeostatic control of both absorption and excretion. Daily diet is the principal source for the body to take in Mn[5]. However, overexposure from environmental sources may lead to elevated Mn level in the body through inhalation or skin penetration. Apart from the small but adequate amount of Mn required for physiological functions, excess Mn is excreted through the biliary, pancreatic, and urinary pathway. Biliary secretion is the main pathway for Mn excretion. As blood passes through the liver, excess Mn is sequestered by hepatocytes and conjugated to bile, then excreted into the intestine[1]. The hepatocytes express several Mn importers, including ferrous ion membrane transport protein (DMT1), transferrin/transferrin receptor (Tf/TfR),

[1]National Key Laboratory of Immunity & Inflammation and Department of Pathophysiology, Naval Medical University, Shanghai, China. [2]NHC Key Laboratory of Hormones and Development, Tianjin Key Laboratory of Metabolic Diseases, Chu Hsien-I Memorial Hospital & Tianjin Institute of Endocrinology, Tianjin Medical University, Tianjin, China. [3]These authors contributed equally: Jian-Hui Shi, Yu-Xia Chen, Yingying Feng, Xiaohang Yang. ✉e-mail: wzhang@smmu.edu.cn

SLC39A8, SLC39A14 and SLC13A5, which regulate Mn influx. Meanwhile, Mn exporters including SLC30A10, SLC40A1 and SPCA1 (Atp2c1) are also expressed by hepatocytes, regulating efflux of excess Mn[1,6].

SLC30A10 deficiency, the first reported inherited disease of Mn excess, was identified in 2012, which provides a key approach to understanding molecular mechanisms of Mn homeostasis[7–9]. As a Mn efflux transporter localizing on the cell membrane, SLC30A10 belongs to the cation diffusion facilitator superfamily of ion transporters, and is mainly expressed in liver, intestine, and brain[10]. Note that ~80% of the body burden of Mn is excreted by the liver, and the intestine excretes the remaining ~20%[11,12]. Loss of function studies revealed that SLC30A10 is essential for Mn excretion by hepatocytes and enterocytes and could be an effective target for pharmacological intervention to treat Mn toxicity[13,14]. Recently, it has been reported that elevated Mn levels could up-regulate *Slc30a10* in liver from 129S4/SvJaeJ mice and small intestine from C57BL/6 J mice mainly through activating HIF1 and HIF2[15]. Although the function of SLC30A10 have been identified, little is known about how the expression of this transporter is regulated, especially at the transcriptional level.

Fructose is commercially added into beverages mainly in the forms of corn syrups for its high relative sweetness. High fructose consumption is perceived as a culprit in metabolic disease. Epidemiological studies indicate a strong correlation between high fructose overconsumption and obesity, non-alcoholic fatty liver disease, type 2 diabetes, kidney dysfunction, and cardiovascular disease[16–18]. Carbohydrate-responsive element binding protein (ChREBP, also known as MLXIPL) is a transcription factor expressed by relevant metabolic tissues[19], and plays critical roles in maintaining glucose and lipid metabolism. It regulates the expression of metabolic genes involved in glycolysis, lipogenesis, fructolysis, and gluconeogenesis[19,20]. Liver-specific ChREBP knockout prevents sucrose- and fructose-mediated induction of de novo lipogenesis

(DNL) enzymes and DNL activity[21,22], and ChREBP protects mice from fructose-induced hepatotoxicity[21]. Fructose has been shown to affect some aspects of the bioavailability of iron, copper, and zinc[23–25], but the literature relating to fructose and Mn bioavailability is meager.

In the current study, we demonstrate that fructose overconsumption enhances hepatic Mn excretion through activating ChREBP/Slc30a10 pathway, thereby reducing liver Mn content. As a result, Mn-dependent enzymatic activity of arginase and Mn-SOD is inhibited after fructose overexposure, and ammonia clearance is impaired. This effect could be recapitulated by hepatic overexpression of Slc30a10, while hepatocyte-specific deletion of Slc30a10 or ChREBP led to an increase in liver Mn content as well as its arginase activity, and abolished the regulatory effect of fructose on liver Mn. Thus, we postulate that fructose overconsumption impairs hepatic Mn homeostasis and ammonia disposal through ChREBP/Slc30a10 pathway, and propose fructose dietary restriction for the prevention and treatment of hyperammonemia.

## Results

### Dietary fructose reduces liver manganese content

To investigate the effect of fructose overconsumption on the homeostasis of metallic elements, we first exposed adult male C57BL/6 J mice to a conventional high fructose diet (HFR, Research Diets), which contains 65% fructose. After two weeks of fructose diet exposure, mass spectrometry analysis did not reveal significant change in plasma levels of Ca, Fe, Mn, Cu, Zn, or Ni compared to normal chow diet (NCD)-fed counterparts (Fig. 1a and Supplementary Fig. 1a). Interestingly, the liver from HFR-fed mice exhibited a significant reduction in Mn content, while their Ca, Fe, Cu, Zn, or Ni contents were comparable with those of NCD-fed mice (Fig. 1b and Supplementary Fig. 1b). Moreover, the decline in liver Mn content also occurred in the mice fed

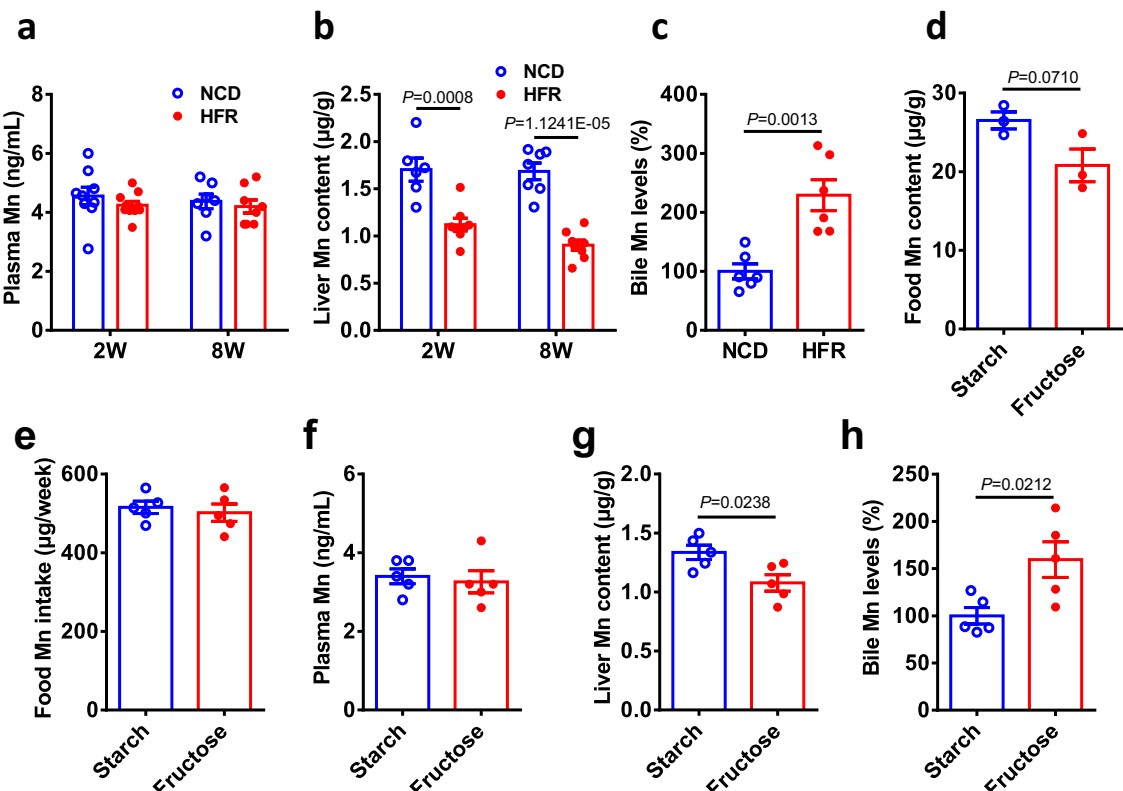

**Fig. 1 | Excessive fructose intake reduces liver manganese content. a**–**c** C57BL/6 J male mice at the age of 2 months were fed normal chow diet (NCD) or a conventional high fructose diet (HFR, 65% fructose, Research Diets) for 2 weeks or 8 weeks (*n* = 6). Their Mn levels in the plasma (**a**), liver (**b**), and bile (**c**). **d**-**h** C57BL/6 J male mice at the age of 2 months were fed 60% starch diet (starch) or 60% fructose diet (fructose) for 1 week (*n* = 5). Food Mn contents (**d**), dietary Mn intake (**e**), Mn contents in the plasma (**f**), liver (**g**), and bile (**h**). Data represent means ± SEM. Two-tailed unpaired Student's *t*-test. Source data are provided as a Source Data file.

HFR for 8 weeks (Fig. 1a, b), while their Zn contents were not changed in the plasma or liver (Supplementary Fig. 1c, d). Of note, two weeks of HFR feeding resulted in a 2.2-fold increase in bile Mn levels (Fig. 1c), implying enhanced Mn excretion from the hepatocytes. Similarly, two weeks' exposure of NCD-fed mice to 10% fructose water did not alter plasma Mn relative to tap water (Supplementary Fig. 2a), but resulted in a remarkable decrease in liver Mn content (Supplementary Fig. 2b), while their Zn levels had no significant change in the plasma or liver (Supplementary Fig. 2c, d).

Considering that the NCD and HFR diets used above have different nutrient components such as protein and fat beyond carbohydrates and Mn per se, we customized a high carbohydrate diet containing 60% corn starch or 60% fructose, respectively, both of which have the same recipe of protein, fat, vitamins, and minerals, but different sources of carbohydrates. Mass spectrometry analysis showed that the 60% fructose diet had a slightly lower content of Mn than the starch diet, without reaching statistical significance (Fig. 1d). During one week of exposure to the 60% fructose diet, C57BL/6 J male mice did not show significant change in body weight compared to control mice on the 60% starch diet (Supplementary Fig. 3a), with a slight increase in the food intake (Supplementary Fig. 3b). As a result, Mn intake from the two different diets was comparable (Fig. 1e). We then characterized Mn distribution in different tissues. These two groups of mice had similar levels of Mn and Zn in the plasma, whole blood, intestine, and kidney (Fig. 1f, and Supplementary Fig. 3c-f). Consistently, liver Mn content was significantly reduced in the mice on the 60% fructose diet compared to the starch counterparts (Fig. 1g), while liver Zn content was similar between the two groups (Supplementary Fig. 3g). In addition, high fructose diet feeding led to a significant elevation in bile Mn concentrations relative to the starch counterparts (Fig. 1h).

To further address whether decreased liver Mn might reflect less efficient absorption of Mn from the intestinal lumen in the presence of fructose, we examined intestinal Mn absorption. Oral gavage with $MnCl_2$ solution resulted in ~100-fold increase in plasma Mn levels in portal vein, with no significant difference between the two groups of mice fed the starch and fructose diet (Supplementary Fig. 4). Collectively, these data suggest that dietary fructose may augment hepatic Mn excretion into the bile, thereby reducing liver Mn content.

## Fructose overconsumption inhibits the activities of Mn-dependent enzymes in the liver

To investigate the biological effect of dietary fructose-induced Mn reduction in the liver, we examined the activities of Mn-dependent enzymes. Arginase is a Mn-dependent enzyme that converts arginine to urea and ornithine, and its activity is positively correlated with Mn concentration within a certain range[26]. Compared to its starch diet control, exposure to the 60% fructose diet led to a mild decrease in liver arginase activity in the mice, and there was a positive correlation between arginase activity and Mn content in the liver from the two groups (Fig. 2a). Superoxide dismutase 2 (SOD2), also known as manganese-dependent superoxide dismutase (Mn-SOD), uses Mn as a cofactor. The mice on the 60% fructose diet showed a decrease in both Mn-SOD activity and total SOD activity in the liver compared with its starch diet-fed counterparts (Fig. 2b, c). Whereas, the protein expression levels of arginase (ARG1), Cu/Zn-SOD, and Mn-SOD were comparable in the liver between the two groups (Fig. 2d). Similarly, compared with NCD, 2 weeks or 8 weeks of HFR feeding also resulted in a significant decrease in the arginase activity, Mn-SOD activity, and total SOD activity in the liver (Fig. 2e-g), and there was a positive correlation between arginase activity and Mn content in the liver, while their protein expression levels of arginase (ARG1), Cu/Zn-SOD, and Mn-SOD were not significantly changed (Fig. 2h). These results indicate that fructose overconsumption does restrain the activity of Mn-dependent enzymes in the liver, at least arginase and Mn-SOD.

## Fructose overconsumption impairs ammonia disposal in injured liver

Given the critical role of liver arginase in ammonia disposal and urea production[27], we then explored whether fructose overconsumption-induced inhibition of liver arginase activity could compromise the clearance of blood ammonia. To this end, we fed C57BL/6 J mice HFR or NCD as the control for 2 weeks before direct challenge by intraperitoneal (i.p.) injection of ammonium chloride[28]. However, there was no significant difference in the clearance of blood ammonia between the two groups (Supplementary Fig. 5a). Considering that patients with hepatic encephalopathy often have impaired ammonia metabolism due to liver dysfunction[29], we further evaluated the potential effect of fructose overconsumption on ammonia disposal in the context of liver dysfunction. Therefore, we induced acute liver injury in NCD or HFR-fed C57BL/6J mice by i.p. injection of carbon tetrachloride ($CCl_4$) (Fig. 2i). After 48 hours of injection of $CCl_4$, both groups of mice displayed a comparable and substantial elevation in plasma levels of transaminase alanine aminotransferase (ALT) and aspartate aminotransferase (AST), which suggests similar liver injury (Supplementary Fig. 5b, c). Of note, the ammonia disposal test revealed a marked impairment of the clearance of blood ammonia in the injured HFR-fed mice compared with their NCD-fed counterparts (Fig. 2j). These results suggest that fructose overconsumption could inhibit arginase activity and consequently impair ammonia disposal at least in the context of liver dysfunction.

## Dietary fructose upregulates Slc30a10 expression in the liver

To explore the molecular mechanism by which fructose overconsumption reduces liver Mn content and Mn-dependent enzyme activity, we characterized the expression pattern of Mn transporters. In consistence with previous reports[30,31], HFR could significantly activate genes encoding ChREBP-β, FGF21, Pklr, Fasn in the liver relative to NCD (Fig. 3a). As a consequence, plasma FGF21 levels were elevated by 2-fold (Fig. 3b). Of note, Slc30a10 mRNA levels were increased by 1.6-fold in the liver compared to their control counterparts (Fig. 3c), while other Mn exporter genes (e.g. Slc40a1 and Atp2c1) or importers genes (e.g. Slc11a2, Slc13a5, Slc39a8, Slc39a14, Trf, and Tfrc) were not significantly changed (Fig. 3c, d). In contrast, Slc30a10 mRNA levels were not affected by HFR in the small intestine (Fig. 3e), despite of the activation of ChREBP-β. Due to the unavailability of antibody against SLC30A10, its endogenous protein expression was not detected. Similarly, 2 weeks of exposure to 10% fructose water resulted in a 1.8-fold increase in Slc30a10 mRNA levels in the liver, together with the activation of ChREBP-β (Fig. 3f). Despite of their identical amount of carbohydrates, one week of exposure of the 60% fructose diet resulted in a marked elevation in the mRNA levels of Slc30a10 as well as ChREBP-β and its glycolytic/lipogenic targets in the liver compared to the starch control diet (Fig. 3g, h), while the mRNA levels of Mn importers and other exporters were similar between the two groups (Supplementary Fig. 6a, b). Western blot revealed that fructose exposure did not significantly alter the expression of HIF1α and HIF2α in the liver relative to the starch control diet (Supplementary Fig. 6c), Consistently, the mRNA levels of Vegf, a target of HIF1, was comparable in the liver between the two groups (Supplementary Fig. 6d). On the other hand, the expression of Slc30a10 mRNA was not changed in small intestine by the 60% fructose diet (Fig. 3i). These results suggest that fructose could specifically upregulate hepatic Slc30a10 expression, thereby enhancing the efflux of intracellular Mn.

## Hepatic Slc30a10 overexpression reduces Mn content and Mn-dependent enzyme activities in the liver

To verify the pathophysiological effect of Slc30a10 upregulation on liver Mn metabolism, we specifically overexpressed Slc30a10 in the liver. To this aim, C57BL/6J mice were administered by intravenous injection with adeno-associated virus (AAV) expressing FLAG-tagged Slc30a10 (AAV-Slc30a10) or the mock control (AAV-Control) at different doses. Overexpression of FLAG-Slc30a10 was confirmed at the

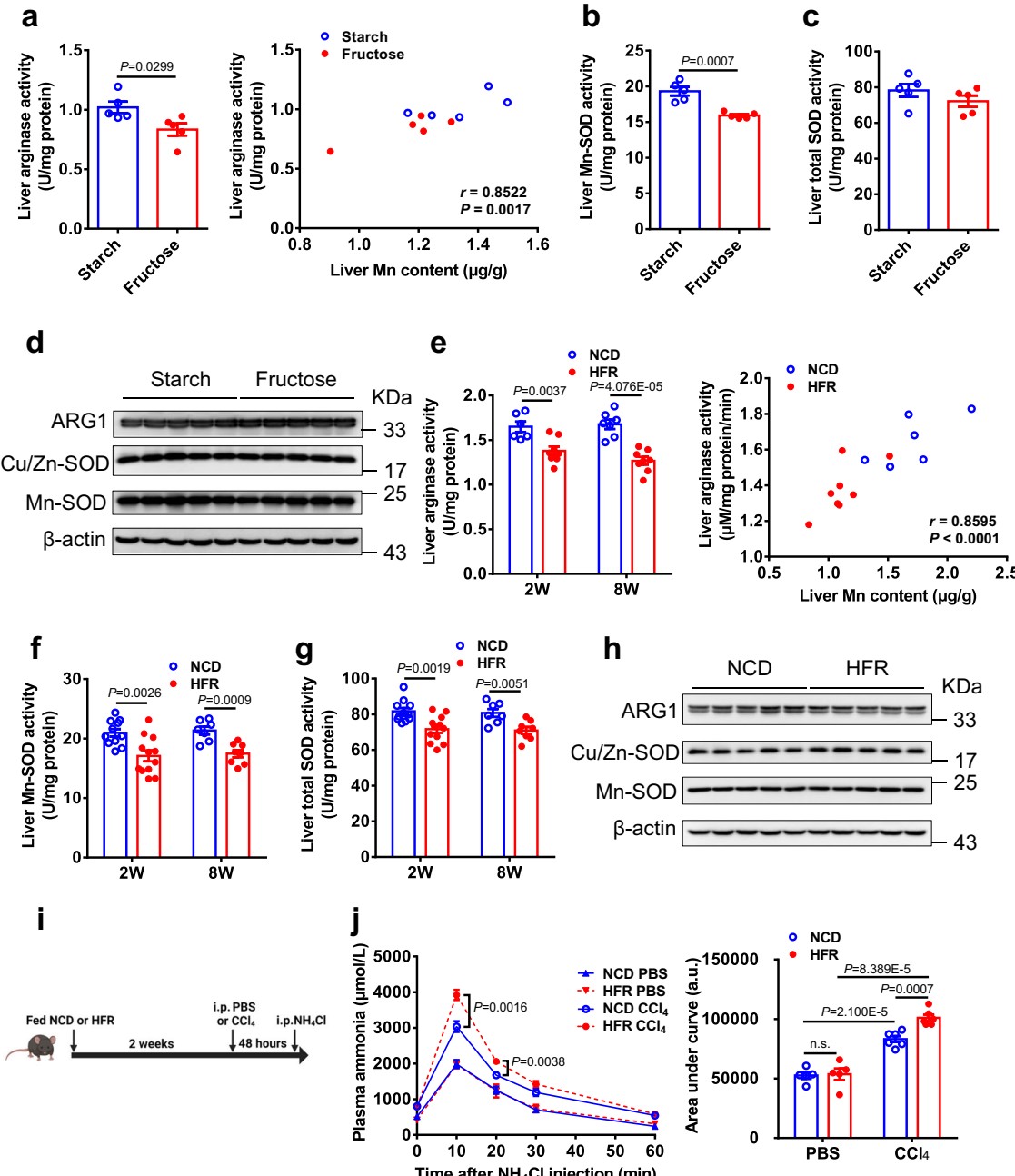

**Fig. 2 | Fructose overconsumption inhibits the activities of Mn-dependent enzymes in the liver and impairs ammonia disposal. a**–**d** C57BL/6 J male mice at the age of 2 months were fed 60% high starch diet (starch) or 60% high fructose diet (fructose) for 1 week (*n* = 5). Liver arginase activity and its correlation with Mn content (**a**), liver Mn-SOD activity (**b**) and total SOD activity (**c**), and protein expression of ARG1, Cu/Zn-SOD, and Mn-SOD in the liver whole lysate analyzed by Western blot with β-actin as a loading control (**d**). **e**–**h** C57BL/6 J male mice at the age of 2 months were fed normal chow diet (NCD) or high fructose diet (HFR, 65% fructose) for 2 weeks or 8 weeks (*n* = 6 ~ 9). Liver arginase activity and its correlation with Mn content for the mice fed NCD or HFR for 2 weeks (**e**) (*n* = 6 ~ 8),

liver Mn-SOD activity (**f**) (*n* = 7 ~ 12) and total SOD activity (**g**) (*n* = 7 ~ 12), and the expression of ARG1, Cu/Zn-SOD, and Mn-SOD protein in the whole liver lysate determined by Western blot for the mice fed NCD or HFR for 2 weeks (**h**). **i** The figure was created with Biorender.com. **j** NCD-fed or HFR-fed mice were i.p. injected with $CCl_4$ (1 ml/kg body weight) or PBS (*n* = 5 ~ 6). Two days later, plasma ammonia levels at the indicated time points after i.p. injection of $NH_4Cl$ (4 mmol/kg body weight) with area under curve (**j**). Data represent means ± SEM. Two-tailed unpaired Student's *t*-test. n.s. denotes not significant. Source data are provided as a Source Data file.

mRNA and protein levels in the liver from AAV-Slc30a10-treated mice by RT-PCR and Western blotting with the anti-FLAG antibody (Fig. 4a, b), respectively. Immunohistochemical staining with anti-FLAG antibody showed that Slc30a10 was expressed in canalicular membranes of hepatocytes (Fig. 4c), colocalized with multidrug-resistance P-glycoprotein 1 (MDR1), which is a marker of canalicular membranes and involved in bile excretion[32]. After 4 weeks of AAV

administration, Slc30a10-overexpressing mice on normal chow showed no significant abnormality in body weight, blood glucose, or the ratio of liver weight to body weight compared with control mice (Supplementary Fig. 7a-c). Most strikingly, overexpression of Slc30a10 led to 50% and 87% reduction in liver Mn content at the low and high dosages (Fig. 4d), respectively, while liver Zn content was unaltered (Supplementary Fig. 7d). As a result, the arginase activity was decreased by

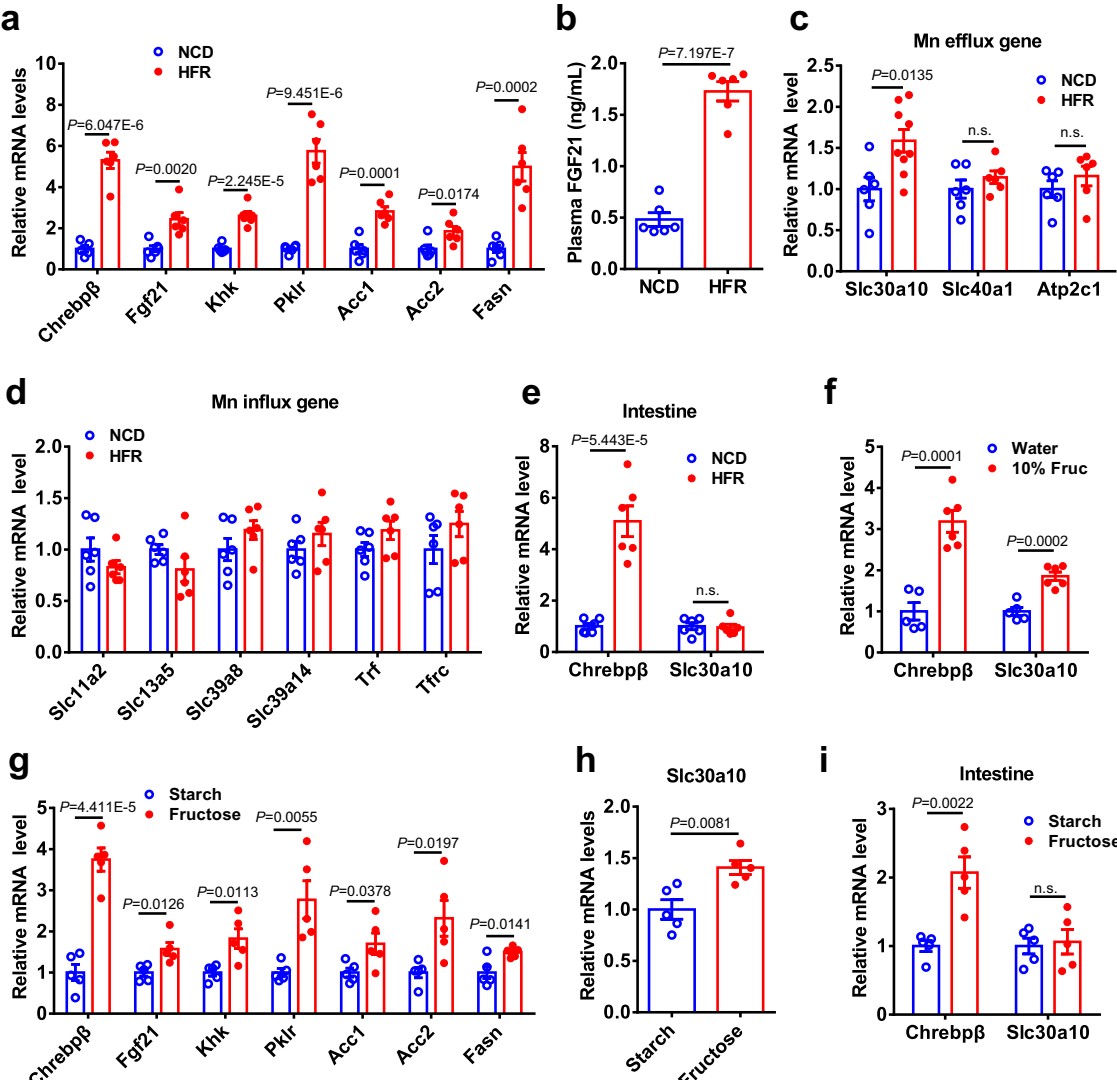

**Fig. 3 | Dietary fructose upregulates Slc30a10 expression in the liver.**
**a**–**e** C57BL/6 J male mice on HFR or NCD for 2 weeks (n = 5 ~ 9). Liver mRNA levels of ChREBP-β and its targets (**a**) (n = 5 ~ 6) and plasma FGF21 levels (**b**) (n = 6), Liver mRNA levels of Mn exporter (**c**) (n = 6 ~ 9) and importers (**d**) (n = 6). ChREBP-β and Slc30a10 mRNA levels in small intestine (**e**) (n = 6). **f** C57BL/6 J male mice on chow diet after exposure to 10% fructose water 2 weeks. Liver mRNA levels of ChREBP-β and Slc30a10 (n = 5 ~ 6). **g**–**i** C57BL/6 J male mice were fed 60% starch or 60% fructose diets for 1 week (n = 5). Liver mRNA levels of ChREBP-β and its targets in glucose and lipid metabolism (**g**) and Slc30a10 (**h**), and intestinal mRNA levels ChREBP-β and Slc30a10 (**i**). Data represent means ± SEM. Two-tailed unpaired Student's t-test. n.s. denotes not significant. Source data are provided as a Source Data file.

22% ~ 50% in the liver by Slc30a10 overexpression (Fig. 4e). Similarly, Mn-SOD activity, and to some lesser extent, total SOD activity was significantly decreased in the liver by the overexpression of Slc30a10 (Fig. 4f, g), while the protein levels of ARG1, Cu/Zn-SOD, and Mn-SOD were unaffected (Fig. 4b). Plasma transaminase ALT and AST levels were slightly but significantly elevated by Slc30a10 overexpression in high dose groups (Supplementary Fig. 7e), however, HE staining and oil red staining of liver sections did not show any obvious morphological abnormality, implying lipid metabolism was largely unaffected in the liver (Supplementary Fig. 7f). Of note, low-dose Slc30a10 over-expression impaired ammonia clearance under the circumstance of $CCl_4$-induced liver injury (Fig. 4h, i). These data suggest that upregulation of hepatic Slc30a10 is sufficient to reduce liver Mn content, thereby impairing the activities of Mn-dependent enzymes.

## Hepatic Slc30a10 is required for fructose to regulate liver Mn homeostasis

To further determine whether Slc30a10 is essential for excess fructose to disrupt hepatic Mn homeostasis, we generated liver-specific Slc30a10 knockout mice (A10-LO) using a Cre-LoxP approach (Supplementary Fig. 8a). To this end, the third exon and the encoding region of the fourth exon of *Slc30a10* gene were flanked with the LoxP sites, and Slac30a10-floxed mice were crossed with albumin promoter-driven Cre transgenic mice. The deletion efficiency of Slc30a10 was confirmed in the liver from A10-LO mice by qRT-PCR analysis (Fig. 5a), suggesting Slc30a10 was predominantly expressed by hepatocytes in the liver. When on normal chow, A10-LO mice did not exhibit significant gross abnormality. Of note, they showed a 1.8-fold elevation in plasma Mn levels as well as 4-fold increase in liver Mn content (Fig. 5b, c), while Zn content was not changed either in the plasma or liver (Supplementary Fig. 8b, c). As a result, deletion of hepatic Slc30a10 led to a 2.3-fold elevation in liver arginase activity, however, liver Mn-SOD or total SOD activity was not affected (Fig. 5d-f), without the expression change of ARG1, Cu/Zn-SOD, and Mn-SOD proteins (Fig. 5g). These data suggest a critical role of hepatic Slc30a10 in the homeostasis of hepatic and circulating Mn under physiological condition.

We then fed A10-LO and control mice HFR for 2 weeks. HFR-induced activation of ChREBP-β, FGF21, and Pklr in the liver was

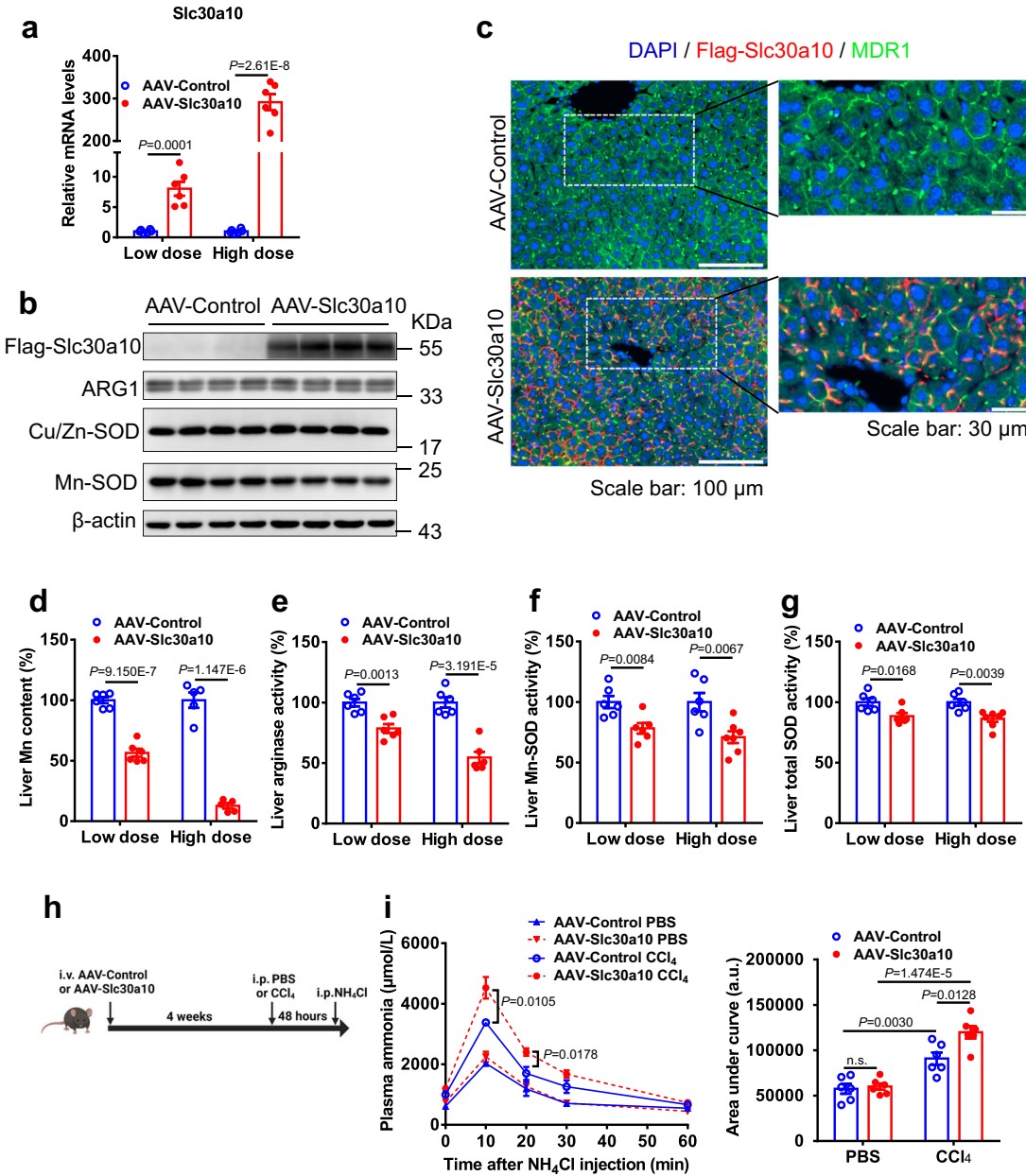

**Fig. 4 | Hepatic overexpression of Slc30a10 reduces Mn content and Mn-dependent enzyme activity. a–g** Two-month-old C57BL/6 J male mice were i.v. injected with AAV expressing Flag-Slc30a10 (AAV-Slc30a10) or AAV-Control at low and high dose ($2\times10^{10}$ vg, and $2\times10^{11}$ vg per mouse, respectively), and fed NCD for 4 weeks before sacrificed ($n = 5 - 7$). Slc30a10 mRNA levels in the liver (**a**) ($n = 6$), Western blot analysis of the liver lysate with FLAG-Slc30a10 detected by anti-FLAG antibody, and protein expression of ARG1, Cu/Zn-SOD, and Mn-SOD in the liver whole lysate analyzed by Western blot with β-actin as a loading control (**b**), FLAG-Slc30a10 expression in the hepatocytes revealed by double immunohistochemical staining with anti-FLAG and anti-MDR1 antibodies with the indicated scale bars (**c**), liver Mn content (**d**) ($n = 5 - 6$), the activity of liver arginase (**e**) ($n = 6$), and the activity of Mn-SOD (**f**) ($n = 6 - 7$) and total SOD (**g**) ($n = 6 - 7$) in the liver. **h** The figure was created with Biorender.com. C57BL/6 J male mice on NCD were i.v. injected with AAV ($2\times10^{10}$ vg per mouse) and pretreated with i.p. $CCl_4$ or PBS as control prior to $NH_4Cl$ challenge at the indicated time points (**h**). Plasma ammonia levels at the indicated time points after $NH_4Cl$ injection with area under curve (**i**) ($n = 6$). Data represent means ± SEM. Two-tailed unpaired Student's *t*-test. Source data are provided as a Source Data file.

comparable between the two genotypes (Fig. 5h). Just like NCD-feeding condition, HFR-fed A10-LO mice also exhibited a robust increase in both plasma and liver Mn levels as well as liver arginase activity relative to HFR-fed control mice (Fig. 5b–d), while their liver Mn-SOD and total SOD activity was slightly increased (Fig. 5e, f). Notably, in the absence of Slc30a10, HFR exposure did not significantly alter liver Mn content as well as the activity of arginase, Mn-SOD, and total SOD in the liver (Fig. 5c–f), while the protein levels of ARG1, Cu/Zn-SOD and Mn-SOD were not altered (Fig. 5g).

We further examined whether the deficiency of Slc30a10 influenced fructose-induced metabolic pathology. After 10 weeks of

HFR feeding, A10-LO and control mice had similar body weight gain, blood glucose levels, plasma triglyceride (TG) and total cholesterol (TC) levels, liver weight to body weight ratio, and liver TG and TC contents (Supplementary Fig. 8d-h). Liver HE staining did not show significant pathological alterations in the two groups, and Oil red O staining revealed comparably mild lipid accumulation (Fig. 5i), with similar plasma ALT and AST levels (Supplementary Fig. 8i). These data suggest that liver Mn accumulation in the absence of Slc30a10 did not cause obvious pathological change, which is consistent with previous reports from other groups[13,14]. Therefore, Slc30a10 is indispensable for fructose-induced

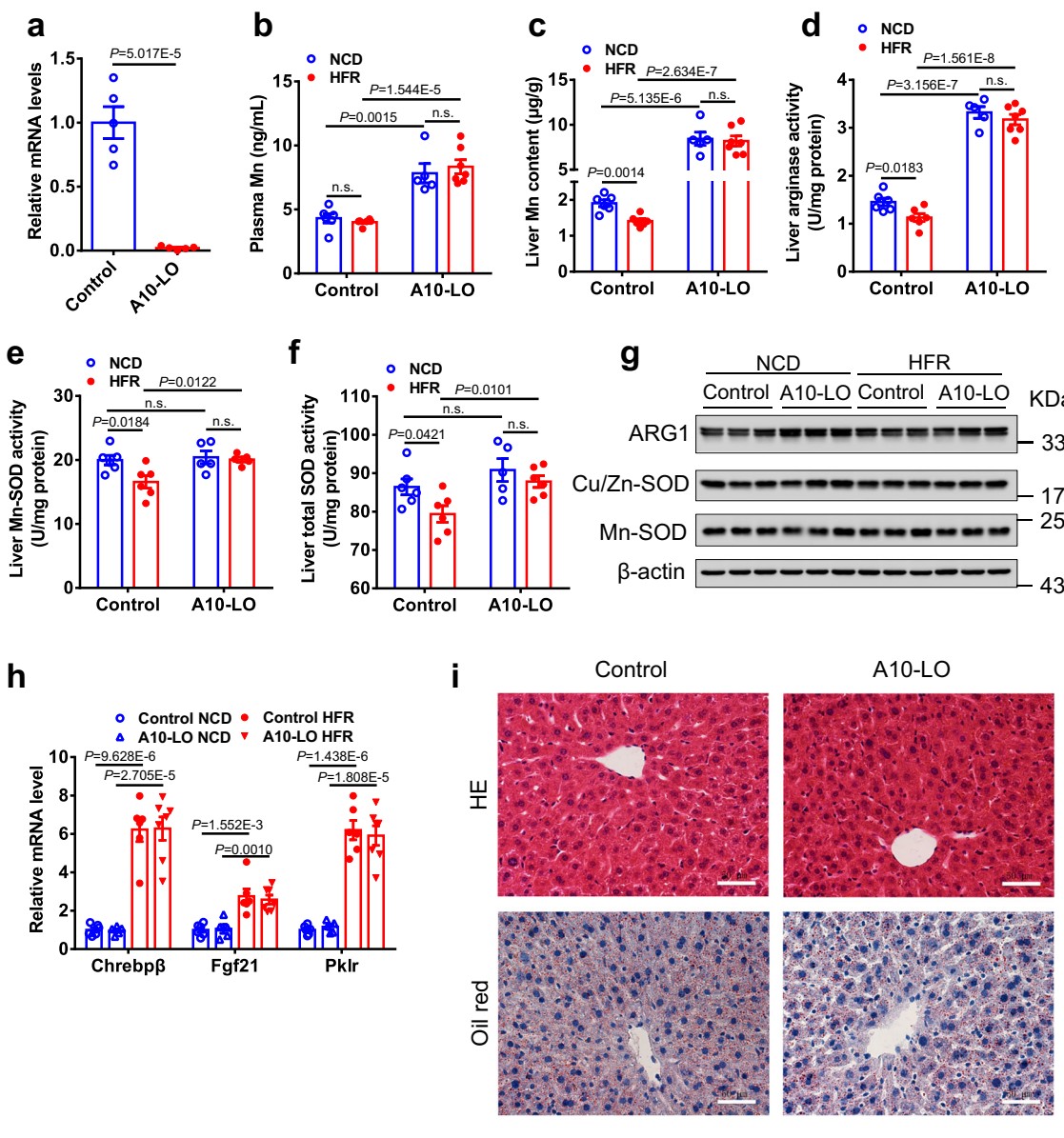

**Fig. 5 | Hepatic Slc30a10 is required for fructose to regulate liver Mn homeostasis. a** Slc30a10 mRNA levels in the liver of control and liver-specific Slc30a10 knockout (A10-LO) mice ($n = 5$). **b-h** Control and A10-LO male mice were fed normal chow diet (NCD) or high fructose diet (HFR) for 2 weeks ($n = 5 \sim 7$). Mn levels in the plasma (**b**) ($n = 5 \sim 7$) and liver (**c**) ($n = 5 \sim 7$), the activity of arginase (**d**) ($n = 5 \sim 7$), Mn-SOD (**e**) ($n = 5 \sim 6$), and total SOD (**f**) ($n = 5 \sim 6$) in the liver, protein expression of ARG1 and SODs in the liver lysate detected by Western blot (**g**), and liver mRNA levels of ChREBP pathway (**h**) ($n = 5 \sim 7$). **i** Representative images for HE staining and Oil red O staining of the liver sections from the mice fed HFR or NCD for 10 weeks. Scale bar: 50 μm. Data represent means ± SEM. Two-tailed unpaired Student's *t*-test. n.s. denotes not significant. Source data are provided as a Source Data file.

reduction of liver Mn, however, has minimal role in glucose and lipid metabolism.

## Hepatic ChREBP is required for fructose to regulate liver Slc30a10 expression and Mn metabolism

Considering that ChREBP is a fructose-responsive transcription factor and mediates fructose-induced biological effects such as de novo lipogenesis (DNL) and gluconeogenesis[21,22], we then explored whether ChREBP is also essential for fructose to regulate liver Mn metabolism. To this aim, we took advantage of the *Chrebp* liver-specific knockout mice (ChLO) which were previously described[21]. Intriguingly, deletion of hepatic ChREBP did not change plasma Mn levels, but led to a marked increase in liver Mn contents under both NCD and HFR feeding conditions (Fig. 6a, b), while liver Zn

content was not changed (Fig. 6c). More importantly, unlike control mice, ChLO mice did not display a significant change in liver Mn content after 2 weeks of fructose exposure compared to NCD-fed counterparts. Consistently, in the absence of hepatic ChREBP, the activities of arginase, Mn-SOD and total SOD were significantly increased in the liver under NCD or HFR-fed conditions, and resistant to the regulatory effect of fructose exposure (Fig. 6d-f), while the protein levels of ARG1, Cu/Zn-SOD and Mn-SOD were not altered (Fig. 6g). On the other hand, plasma or liver Zn contents were similar between ChLO and control mice under the different feeding conditions (Fig. 6c, and Supplementary Fig. 9a). These data strongly suggest an essential role of ChREBP in the regulation of hepatic Mn content and Mn-dependent enzyme activity by dietary fructose.

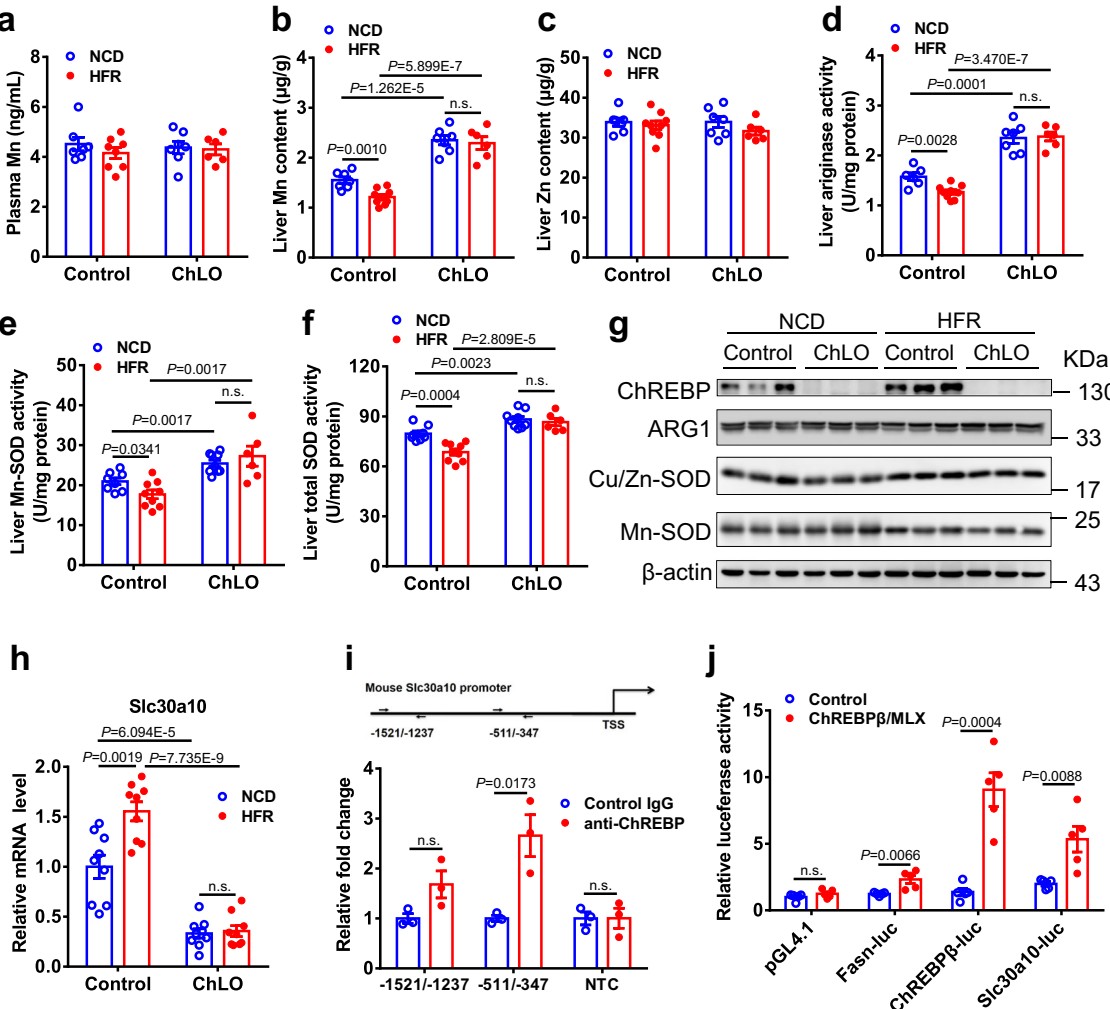

**Fig. 6 | Hepatic ChREBP is required for the regulation of liver Slc30a10 expression and Mn homeostasis by fructose. a–h** Male control and ChREBP liver-specific knockout (ChLO) mice were fed NCD or HFR for 2 weeks (*n* = 6 - 9). Plasma Mn levels (**a**) (*n* = 6 - 8), liver Mn (**b**) (*n* = 6 - 9) and Zn (**c**) (*n* = 6 - 9) contents, liver arginase activity (**d**) (*n* = 6 - 9), liver Mn-SOD (**e**) (*n* = 6 - 9) and total SOD (**f**) (*n* = 6 - 9) activity, Western blot analysis for ChREBP, ARG1, and SODs in the liver lysate (**g**), and liver mRNA expression of Slc30a10 (**h**) (*n* = 9). **i** ChIP analysis revealed the binding of ChREBP to Slc30a10 promoter in the liver, with the primers indicated by arrows at the position upstream the transcriptional start site (TSS). The experiments were independently repeated 3 times. NTC, negative control primer. **j** Overexpression of ChREBP-β/MLX enhanced the transcriptional activity of the Slc30a10 promoter in 293 T cells. The reporters Fasn-luc and ChREBPβ-luc were used as positive controls. The experiments were independently repeated 5 times. Data represent means ± SEM. Two-tailed unpaired Student's *t*-test. n.s. denotes not significant. Source data are provided as a Source Data file.

### ChREBP is a transcriptional activator of Slc30a10

Considering the similar effects of hepatic ChREBP and Slc30a10 on liver Mn homeostasis and their responsiveness to fructose stimulation, we speculated that ChREBP might regulate Mn metabolism through Slc30a10. To test this hypothesis, we characterized the expression patterns of genes involved in Mn transportation in ChLO mice. ChREBP deletion led to a 50% reduction in Slc30a10 mRNA levels, without significant changes of the mRNA levels of Mn importers (e.g. Slc11a2, Slc13a5, Slc39a8, Slc39a14, Trf, and Tfrc) or other Mn exporters (*Slc40a1* and *Atp2c1*) (Supplementary Fig. 9b). Importantly, fructose lost activating effects on liver *Slc30a10* in the absence of ChREBP (Fig. 6h). These results suggest that ChREBP is required for the regulation of *Slc30a10* gene in liver by fructose.

To further determine whether *Slc30a10* is a direct target gene of ChREBP, we first examined the binding of ChREBP protein to the *Slc30a10* gene. Chromatin immunoprecipitation (ChIP) analysis revealed ChREBP occupancy onto the promoter of Slc30a10 gene in liver, with a 2.6-fold enrichment onto −511/−347 (Fig. 6i). We then constructed a luciferase reporter harboring the promoter of Slc30a10

(−1520/+130, Slc30a10-Luc), and examined its activity in 293 T cells. Cotransfection with the plasmids expressing ChREBPβ and its obligate partner MLX resulted in a 2.7-fold increase in the activity of the reporter Slc30a10-luc. As positive controls, the promoter reporter of Fasn-Luc and ChREBPβ-Luc were also activated by ChREBPβ/Mlx by 1.9-fold and 6.6-fold, respectively (Fig. 6j). Collectively, these data suggest that ChREBP could bind to the promoter and act as a transcriptional activator of the Slc30a10 gene, thereby mediating the regulation of hepatic Mn homeostasis and ammonia disposal by dietary fructose in male mice (Fig. 7).

### Discussion

Dietary sugars have been shown to affect the bioavailability of iron, zinc, copper[23,24,33–35]. However, there is limited literature linking fructose to Mn metabolism. There is a previous observation that fructose-fed subjects had higher fecal excretions of manganese, iron, calcium, phosphorus, and magnesium than the control subjects who were on a self-selected diet[23], which was largely attributed to their intestinal malabsorption due to diarrhea. The current study establishes the

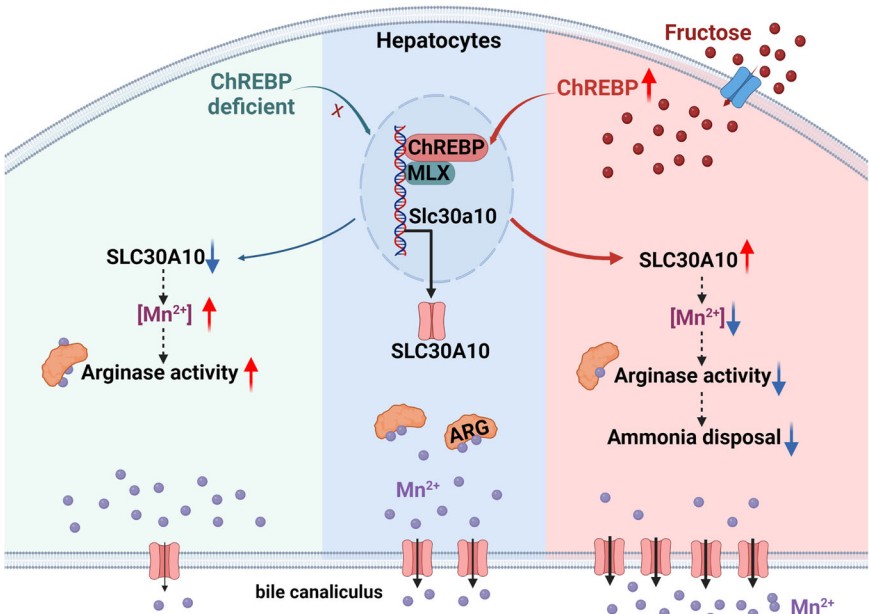

**Fig. 7 | Schematic illustration for the regulation of hepatic Mn homeostasis and ammonia disposal by fructose overconsumption.** The Mn exporter Slc30a10 is a target of ChREBP in liver. In the absence of ChREBP, Slc30a10 expression is reduced, and intracellular Mn contents as well as arginase activity are increased. Dietary fructose activates the transcription of Slc30a10 through ChREBP, thereby promoting Mn efflux and reducing Mn content. ARG, arginase. The figure was created with Biorender.com.

effect of fructose overconsumption on liver Mn homeostasis through ChREBP/Slc30a10 pathway (Fig. 7). First, the decline of liver Mn content was consistently caused by short-term (1 ~ 2 weeks) or long-term (8 ~ 10 weeks) overexposure to fructose in the form of high fructose diet vs NCD or fructose solution vs tap water. Typically, this effect was recapitulated in the mice fed the 60% fructose diet vs the starch control diet, the two groups of which had no significant difference of dietary Mn intake, intestinal Mn absorption, and Mn contents in blood, plasma, kidney, and intestine. Consistently, bile Mn levels were elevated by fructose overexposure, indicative of enhanced Mn excretion from hepatocytes. Meanwhile, Zn contents in the liver and other extrahepatic tissues were not affected. As a result, the activity of Mn-dependent enzymes (e.g. arginase and Mn-SOD) was increased in the liver by fructose overconsumption, and there was a correlation between liver Mn contents and arginase activity. Second, fructose overconsumption activates the expression of Slc30a10 specifically in the liver but not in intestine, without affecting other Mn transport genes. Slc30a10 overexpression can largely mimic the effects of fructose overconsumption on liver Mn metabolism, including the decreased liver Mn content and enzymatic activity of arginase and Mn-SOD. Third, deletion of hepatic Slc30a10 or ChREBP could abolish the regulatory effect of fructose on liver Mn homeostasis and Mn-dependent enzyme activities.

This study also established the critical role of ChREBP in the regulation of liver Mn homeostasis through Slc30a10. ChREBP is one of the major transcription factors regulating carbohydrate metabolism and lipogenesis[21,30,36,37]. The expression and activity of ChREBP are induced by carbohydrates, especially fructose, which is relevant in the pathogenesis of metabolic diseases caused by excessive fructose consumption[38,39]. We have successfully identified Slc30a10 as a novel target of ChREBP. Liver-specific deletion of ChREBP led to a marked decrease in the basal levels of Slc30a10 mRNA expression under normal feeding condition, and more importantly, completely abolished the regulatory effect of fructose on Slc30a10 gene expression as well as Mn homeostasis in the liver. Whereas the protein expression of HIF1 and HIF2, which could up-regulate SLC30A10 in liver and small intestine[15], was not affected in the liver by fructose overexposure. The

evidence strongly supports that ChREBP is essential for the activation of liver Slc30a10 by fructose. Biochemically, ChIP analysis and dual luciferase reporter assay reveal that ChREBP can directly bind to and transcriptionally activate Slc30a10 gene promoter. We failed to identify the typical ChoRE sequence usually present in other target genes of ChREBP[37,40], suggesting some undefined sequence or motif other than classical ChoRE may mediate the binding of ChREBP onto the *Slc30a10* gene. On the other hand, SLC30A10 is also involved in intestinal Mn excretion, which is critical for systemic Mn homeostasis when hepatic Mn excretion is compromised or saturated[13,14]. Vitamin D3 and bile acids were reported to induce SLC30A10 expression in intestine[41–43]. Our data reveal the intestinal *Slc30a10* gene is not regulated by fructose, the mechanism of which is unclear. Therefore, studying the crosstalk between fructose/ChREBP, Mn/HIF, vitamin D3, and bile acid-mediated regulation of *Slc30a10* will facilitate to obtain a comprehensive understanding of Mn homeostasis.

Our findings also raise a concern about the deleterious effect of dietary fructose on ammonia disposal particularly under liver dysfunction. Ammonia is mainly derived from intestinal deamination and is disposed by conversion into urea through the urea cycle in liver, in which arginase catalyzes the conversion of L-arginine to L-ornithine and urea. In the case of liver failure, ammonia accumulates and shunts into the systemic circulation. Hyperammonemia leads to neuronal dysfunction and hepatic encephalopathy[28]. Patients with hepatic encephalopathy generally need to limit protein intake, and lactulose is the most commonly used non-absorbable disaccharide for the treatment of hepatic encephalopathy. Here we show that fructose overconsumption could inhibit the clearance of circulating ammonia in mice with liver dysfunction due to impaired arginase activity (Fig. 7). Therefore, in addition to limiting protein intake, patients with hyperammonemia may also need to restrict fructose consumption.

Our work also helps to elucidate the mechanism about regulation of SOD activity by fructose. Previous studies have reported that fructose intake can reduce SOD activity[44–46], but the mechanism is not well understood. Here we show that fructose intake can reduce Mn-SOD activity and total SOD activity by reducing liver Mn content, which provides a new perspective for studying the relationship between

fructose intake and SOD activity. Considering liver-specific Mn-SOD knockout mice showed no obvious morphological abnormalities or biochemical alterations in the liver[47,48], the effect of fructose intake on Mn-SOD activity may be reflected in some pathological states such as fatty liver. Interestingly, although fructose intake reduced liver SOD activity by reducing liver Mn content, we did not observe elevated Mn-SOD activity in the liver from Slc30a10-deficient mice despite the presence of substantially elevated Mn content, probably due to the saturation of Mn-SOD by Mn. By contrast, ChLO liver showed a significant increase in both SOD activity and Mn content. This discrepancy implicates an alternative mechanism beyond Slc30a10 about the regulation of SOD activity by ChREBP, which merits further investigation.

Collectively, our findings point out that fructose overconsumption impairs liver Mn homeostasis and ammonia disposal through ChREBP/Slc30a10 pathway, and provide caution for excess fructose intake under liver dysfunction.

## Methods

### Animal studies

Liver-specific ChREBP knockout (ChLO) mice were generated as previously described[21]. Briefly, an embryonic stem cell-based strategy was used to generate ChREBP-floxed mice by flanking the 8th exon with LoxP sites, which were crossed with *Alb-Cre* transgenic mice to generate ChLO mice. Embryonic stem cells with the *Slc30a10* gene targeted by knock-first strategy were obtained from CAM-SU Genomic Resource Center (www.cam-su.org), the 3rd and 4th exons of which were flanked with LoxP sites. *Slc3010*-floxed mice were obtained by crossing the founder onto *Flp* transgenic mice[49], which resulted in deletion of the LacZ-Neo cassette between flippase recognition targets (FRTs). Slc30a10 liver-specific knockout (A10-LO) mice were generated by crossing the Slc30a10$^{flox}$ mice with albumin-Cre transgenic mice. All mice were maintained on a 12-h light-dark cycle in a temperature controlled at 25°C with 40%-60% humidity and given *ad libitum* access to food and water unless otherwise stated. PCR primers for genotyping are listed in Supplementary Table 1.

All animal experiments were performed on mice that were backcrossed onto C57BL/6 J for at least six generations and approved by the Naval Medical University Animal Ethics Committee (Shanghai, China). The mice were fed a purified AIN-93G diet. High fructose diet containing 65% fructose in calorie were purchased from Research Diets (D11707R). The 60% starch diet and 60% fructose diet were synthesized by Shanghai Fanbo Biotechnology Company. In the intestinal Mn absorption experiment, the mice were administered by oral gavage with manganese chloride (72 mg/kg MnCl$_2$ • 4H$_2$O) or equal volume saline. Thirty minutes later, portal vein blood was collected in the presence of EDTA as an anticoagulant, and the plasma was separated for Mn detection. Adult male mice were used for the experiments unless indicated otherwise.

### Metabolic Assays

We measured blood glucose using a glucose monitor (One Touch Ultra, Lifescan; Johnson & Johnson, Milpitas, CA), and measured liver TG by colorimetric assays (Sigma and Wako)[50]. We extracted TG from liver with Trion X-100 for colorimetric assays (Sigma, TR0100). Plasma aminotransferases were measured in a biochemical analyzer (Mindray BS-240) with the kits provided by the manufacturer (Mindray, Shenzhen, China). We detected liver arginase activity with the arginase activity assay kit (Sigma, MAK112) following operation manual guidelines. Liver Mn-SOD activity was detected according to the protocol of Mn-SOD Assay Kit (Beyotime Biotechnology, S0103). Liver total SOD activity was detected according to the protocol of Total SOD Assay Kit (Beyotime Biotechnology, S0101). Plasma ammonia was detected with a Blood Ammonia Analyzer (Arkray, PocketChem BA, PA-4140) following operation manual guidelines.

Metallic elements were detected using ICP-MS (PerkinElmer, NexION 2000B) system at Shanghai Xinhua Hospital and WEIPU Chemical Technology Service Company. Liver samples were weighted, digested with 6 ml HNO$_3$/HCl [3:1] at 70°C overnight, and diluted to a final volume of 15 ml for analysis. The metallic elements were measured using a calibration curve of an aqueous standards prepared at 5 different concentrations of each metal. The results were reported in p.p.m. based on volume. The total amount, determined by multiplying the concentration with the volume, was divided by the wet weight of the liver tissue to obtain the metal content per gram of the liver. Mn concentrations in the bile were normalized to liver mass.

### mRNA Expression Analysis

mRNA expression in the liver and jejunum was measured by quantitative RT-PCR (qRT-PCR) using the SYBR Green dye-based assay with the *36B4* gene as internal control in every plate[51]. The quantitative PCR primer sequences are listed in the Supplementary Table 2.

### Protein Expression Analysis

Protein expression in the liver were detected as described previously[51]. Briefly, whole liver tissue lysates were generated using urea lysis buffer and run on 10% SDS-PAGE gel. Proteins were transferred onto polyvinylidene fluoride membranes (PolyScreen) and incubated in sequential order with the appropriate primary antibody and horseradish peroxidase–conjugated secondary antibodies. This was followed by visualization with enhanced chemiluminescence reagents (Pierce Biotechnology, Rockford, IL). The antibody information is listed in Supplementary Table 3. Uncropped, unprocessed scans of the blots are available in the Source Data file.

### Histology and immunohistochemical staining

Pieces of the liver fixed in 4% paraformaldehyde were embedded in paraffin and stained with hematoxylin and eosin. For Oil Red O staining, we froze liver tissues in OCT compounds before cutting at 8 μm, and performed staining according to standard protocol[50]. Double immunohistochemical staining were performed with paraffin-embedded liver sections as described previously[52], with some modifications. Briefly, the sections were incubated with mouse anti-FLAG monoclonal antibody and rabbit anti-MDR1 antibodies, and visualized with the indirectly coupled Alex-594 and Alex-488, respectively.

### Administration of adeno-associated virus

FLAG-tagged human Slc30a10 cDNA from Mukhopadhyay's laboratory[10] was cloned into the NheI and HindIII restriction sites of the AAV8 vector pAAV-CAG-MCS-3xFLAG-WPRE. The viruses were packaged in HEK293T cells by co-transfected with pAAV2/8-RC and pHelper using PEI MAX 40 K. After 72 hours of transfection, the supernatants were harvested and filtered with 0.22 μm filter, then concentrated with Amicon Ultra-15 centrifugal filter unit, the titers of virus stock were determined by qPCR. C57BL/6 J male mice were administered single tail vein injection of 200 μl PBS containing $2 \times 10^{10}$ (low dose) or $2 \times 10^{11}$ (high dose) vector genomes of AAV-Slc30a10 or AAV-control, and analyzed 4 weeks later.

### Chromatin Immunoprecipitation Assay

Liver tissue was isolated from normal adult mice and flash-frozen in liquid nitrogen. ChIP assays of liver tissue were performed as previously described with some modifications[50]. Briefly, liver tissue was crosslinked with 1% formaldehyde for 7 min at room temperature before stopped by adding 0.5 mol/L glycine. After sonication, fragmented chromatin was incubated overnight at 4°C with anti-ChREBP antibody (Novus, NB400-135), or isotype IgG (Upstate) as negative control, and followed by incubation with Dynabead-conjugated Protein G (Invitrogen). Purified chromatin DNA was subjected to real-time

PCR analysis with the primers of the *Slc30a10* gene listed in Supplementary Table 4.

## Reporter Assay

Mouse ChREBP-β cDNA was cloned by PCR using ChREBP α-expressing plasmid (a gift from Dr. Catherine Postic) as template[53], and MLX cDNA was cloned by RT-PCR from the liver, both of which were introduced into the expression vector pCMV-tag2 (Stratagene). The mouse *Slc30a10* gene promoter (−1520/+130) was cloned by PCR using genomic DNA as template, with forward primer 5′CCCTCGAGCTCCCCCATCCACACCCCACTTG3′, and reverse primer 5′CGGGATCCTTACCCGCCCGGACCTTTTATGCT3′. The mouse *Fasn* promoter (−245 bp) was purchased from Addgene and then inserted into pGL4.1-basic vector (Promega). All the plasmids were confirmed by DNA sequencing. The reporter assay was performed as described previously[50]. Briefly, human embryonic kidney cells 293 T (ATCC, CRL-3216) were transfected with plasmids using Lipofectamine 3000, and the luciferase activity was detected 48 h after transfection using Dual-luciferase assay kit (Promega) and normalized by the internal control RL-TK.

## Statistical Analysis

All numerical data were expressed as mean ± SEM. Statistically significant differences among the means of different groups were determined by a two-tailed unpaired Student's *t*-test and defined as $P < 0.05$.

## Reporting summary

Further information on research design is available in the Nature Portfolio Reporting Summary linked to this article.

# Data availability

All data generated or analyzed during this study are included in this article (and its supplementary information files). Source data are provided with this paper.

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

## Acknowledgements
This work was supported by the grants from National Key R&D Program and National Natural Science Foundation of China (2019YFA0802500 to W.J.Z., 2018YFA0800602 to W.J.Z., 91857203 to W.J.Z., U23A20171 to W.J.Z., 31730042 to W.J.Z., 82000834 to J.H.S.); Collaborative Innovation Program of Shanghai Municipal Health Commission (2020CXJQ01 to W.J.Z.). We thank Qianlong Zhang and Jun-Xia Liu for technical assistance with mass spectrometry analysis of metallic elements, and Dr. Somshuvra Mukhopadhyay for providing the plasmid of Slc30a10 cDNA.

## Author contributions
W.J.Z. and J.H.S. conceived the project and designed the experiments. J.H.S., Y.F., X.Y., J.L., T.W., C.C.W., X.H.M., R.Y. and D.C. performed experiments. H.Z., X.X. and Z.X. contributed to analyze data. W.J.Z., Y.X.C. and J.H.S. wrote the manuscript. W.J.Z. and J.H.S. obtained funding, and W.J.Z. contributed to study supervision. W.J.Z. is the guarantor of this work and, as such, has full access to all the data in the study and takes responsibility for the integrity of the data and the accuracy of the data analysis.

## Competing interests
The authors declare no competing interests.
