## [Peer Review File · Nature Communications]

Fructose Overconsumption Impairs Hepatic Manganese Homeostasis and Ammonia DisposalEditorial Note: Parts of this Peer Review File have been redacted as indicated to maintain the confidentiality of unpublished data.

REVIEWER COMMENTS

Reviewer #1 (Remarks to the Author):

The current work from Dr. Zhang group investigated role of hepatocyte ChREBP in fructose overconsumption-impaired Manganese (Mn) homeostasis. It is well known about negative metabolic consequences following long-term excessive fructose intake in human physiology but the underlying molecular mechanisms are still not fully understood. In the current study, authors tested whether hepatocyte ChREBP deficiency impacts liver Mn metabolism following high fructose intake in mice. Given the clinical significance of ChREBP in fructose metabolism, this work identifies a novel link between ChREBP and Mn homeostasis upon fructose intake. Both in vivo and in vitro data are strong and data interpretation is appropriate. However, several major limitations are also identified in the current study and need to be addressed with further experiments.

(1) The overall scope of this study is quite narrow and most data were generated from short-term feeding with high fructose diet or fructose water (1 - 2 weeks). It is critical to examine effects of long-term fructose diet feeding on liver Mn homeostasis and Mn-dependent Arginase and Mn-SOD activity. How about female mice?

(2) It has been well known that high fructose diet feeding can stimulate hepatic lipogenesis, FGF21 and fructolysis. Authors should include some of these markers in their study in addition to ChREBP. Furthermore, it is likely that these pathways can affect liver Mn content independent of expression of transporters. This hypothesis was not tested at all in the current study.

(3) Figure 3C, induction of Slc30a10 by fructose diet is less than 1.5 fold but reached almost 3 fold by fructose water. Any explanations? Also protein levels of SLC30A10 should be measured in the same treatments.

(3) AAV-mediated overexpression of Slc30a10 led to super-physiological levels of SLC30A10 in hepatocytes (Figure 4). Authors should dial down the expression levels which are comparable with high fructose diet feeding.

(4) Figure 5, A10-LO is an interesting mouse model to study hepatic Mn homeostasis. Given its strong effect on hepatic Mn content, authors should explore how these mice respond to long-term HFR diet feeding in terms of liver steatosis, liver injury and insulin resistance. Also how female A10-LO mice respond to HFR feeding?

(5) Figure 6, the expression of Slc30a10 should be examined in WT vs ChKO following RC or HFR diet. Also which ChREBP protein was detected in Figure 6F, ChREBPa or ChREBPb? ARG1 levels seem to be downregulated in WT mice fed with NCD. Any explanation?

Reviewer #2 (Remarks to the Author):

In this manuscript, the authors investigate the impact of fructose overconsumption on manganese (Mn) levels and other related parameters. The authors first showed that fructose intake via diet or drinking water lowered liver levels of Mn but not other metals. Activities of Mn-dependent enzymes in the liver were also reduced, although protein levels were unaffected. The authors also show that fructose exposure of mice with carbon tetrachloride-induced liver injury altered plasma ammonia clearance. They also identified that fructose upregulates expression of ChREBP, a transcription factor known as carbohydrate responsive element binding protein, and Slc30a10, a transporter that exports Mn from hepatocytes into bile. Finally, the authors showed that hepatocyte deficiency in ChREBP or Slc30a10 abrogates fructose-induced changes in Mn levels. Overall, their data support a model in which excess fructose consumption leads to increased ChREBP-dependent Slc30a10 expression in the liver, leading to altered liver Mn levels.

The authors present interesting findings and employ multiple approaches to interrogate the link between fructose and manganese. However, we do have suggestions for them on how this manuscript can be improved:

1. Are there changes in Mn levels in organs other than the liver upon fructose feeding?

Measuring Mn levels in other organs would be very informative. It could indicate if Mn levels are decreased at the whole body level (suggesting an imbalance between absorption and excretion) or perhaps Mn is just redistributed from liver to other organs. Also, the authors measured blood Mn levels by focusing on plasma. Hemolysis can falsely elevate plasma/serum Mn levels. Were there changes in whole blood?

2. Were bile Mn levels altered in mice treated with fructose? This would be consistent with the increased Slc30a10 expression observed in fructose-treated mice. Also, did the authors assess Mn absorption from the diet? Could decreased absorption contribute to decreased liver Mn levels? Similarly, could the change in fructose levels in the diets impact the bioavailability of Mn for absorption?

3. Slc30a10 is expressed in the intestines and can export Mn from enterocytes into the GI tract. Are there changes in Slc30a10 expression in the intestines in fructose-treated mice?

4. The experiments with CCl₄ lack a 'no treatment' control. As presented, all mice are treated with CCl₄.

5. For the immunofluorescence presented in Figure 4B, is Slc30a10 expressed in canalicular membranes in hepatocytes? It appears as such, but this could be confirmed using markers such as MDR1.

6. AAV-mediated Slc30a10 overexpression resulted in increased transaminase activity. Did the authors use saline-treated controls? Could AAV treatment on its own alter AST and ALT levels? Also, are there changes in ammonia clearance upon overexpression?

7. For Figure 5B, a statistically significant difference is noted in liver Mn levels in mice on control vs high fructose diets. The levels appear somewhat similar. Can the authors confirm that these groups differ significantly? Perhaps they could use a split y axis so that it is easier to approach the difference in Mn levels in these groups? The same comment applies to panels C and E.

8. The figure panel labels for figure 4 do not align with the descriptions in the legend.

9. The authors noted changes in activity of Mn-dependent enzymes without changes in protein levels. Does the metal status of these proteins not impact the turnover of the proteins? Do Mn levels not impact the expression of these proteins either? Has this been addressed in other papers?

10. A recent paper (cited in the manuscript) has shown that hypoxia-inducible factors also regulate Slc30a10 expression. Does fructose impact HIF-dependent Slc30a10 expression?

Reviewer #1 (Remarks to the Author):

The current work from Dr. Zhang group investigated role of hepatocyte ChREBP in fructose overconsumption-impaired Manganese (Mn) homeostasis. It is well known about negative metabolic consequences following long-term excessive fructose intake in human physiology but the underlying molecular mechanisms are still not fully understood. In the current study, authors tested whether hepatocyte ChREBP deficiency impacts liver Mn metabolism following high fructose intake in mice. Given the clinical significance of ChREBP in fructose metabolism, this work identifies a novel link between ChREBP and Mn homeostasis upon fructose intake. Both in vivo and in vitro data are strong and data interpretation is appropriate. However, several major limitations are also identified in the current study and need to be addressed with further experiments.

(1) The overall scope of this study is quite narrow and most data were generated from short-term feeding with high fructose diet or fructose water (1-2 weeks). It is critical to examine effects of long-term fructose diet feeding on liver Mn homeostasis and Mn-dependent Arginase and Mn-SOD activity. How about female mice?

[Editorial Note: REDACTED]

(2) It has been well known that high fructose diet feeding can stimulate hepatic lipogenesis, FGF21 and fructolysis. Authors should include some of these markers in their study in addition to ChREBP. Furthermore, It is likely that these pathways can affect liver Mn content independent of expression of transporters. This hypothesis was not tested at all in the current study.

Response: According to your suggestion, we have added the data about the other ChREBP targets such as FGF21, and Pklr, Fasn (revised Fig.3a). In liver-specific Slc30a10 knockout mice (A10-LO), we demonstrated that high fructose diet could also activate the expression of FGF21 and other ChREBP targets in the liver (revised Fig.5h), but lost its regulatory effects on liver Mn contents and Mn-dependent arginase and Mn-SOD activity compared to normal chow (revised Fig. 5d,e). Therefore, it is unlikely that FGF21 or lipogenic pathway can affect liver Mn content independently of Slc30a10 expression. At present, we cannot completely exclude the possibilities independent of other transporters than Slc30a10.

(3) Figure 3C, induction of Slc30a10 by fructose diet is less than 1.5 fold but reached almost 3 fold by fructose water. Any explanations? Also protein levels of SLC30A10 should be measured in the same treatments.

Response: We double-checked the data. As shown in Figure 3, induction of liver Slc30a10 is 1.37 fold after 1 week of 60% fructose diet compared to the starch diet, and 1.85 fold by 2 weeks of 10% fructose drink vs tap water. The difference of changing magnitude could be explained by fructose feeding duration and different reference baseline. Currently, there is no commercially available antibody against mouse Slc30a10 working for immunoblot or immunohistology, therefore we can't characterize the expression of endogenous SLC30A10 in mice.

(3) AAV-mediated overexpression of Slc30a10 led to super-physiological levels of SLC30A10 in hepatocytes (Figure 4). Authors should dial down the expression levels which are comparable with high fructose diet feeding.

Response: According to your suggestion, we repeated the Slc30a10 overexpression experiments with low-dose of AAV, which resulted in a 7.5-fold increase in Slc30a10 mRNA levels in the liver. As a result, liver Mn and its related enzymes activities were reduced. The data obtained by low-dose AAV were integrated into the revised Fig.4.

(4) Figure 5, A10-LO is an interesting mouse model to study hepatic Mn homeostasis. Given its strong effect on hepatic Mn content, authors should explore how these mice responds to long-term HFR diet feeding in terms of liver steatosis, liver injury and insulin resistance. Also how female A10-LO mice respond to HFR feeding?

[Editorial Note: REDACTED]

(5) Figure 6, the expression of Slc30a10 should examined in WT vs ChLO following RC or HFR diet. Also which ChREBP protein was detected in Figure 6F, ChREBP_a or ChREBP_b? ARG1 levels seems to downregulated in WT mice fed with NCD. Any explanation?

Response: Thank you for your helpful suggestion, and we have included the Slc30a10 expression in ChLO liver after excessive fructose exposure (Fig.6h). The ChREBP detected by Western blot was ChREBP_a, which has been added. We double-checked all the ARG1 blots and repeated some samples, and found no significant changes, and replaced with a more representative blot (Fig.6g).

Reviewer #2 (Remarks to the Author):

In this manuscript, the authors investigate the impact of fructose overconsumption on manganese (Mn) levels and other related parameters. The authors first showed that fructose intake via diet or drinking water lowered liver levels of Mn but not other metals. Activities of Mn-dependent enzymes in the liver were also reduced, although protein levels were unaffected. The authors also show that fructose exposure of mice with carbon tetrachloride-induced liver injury altered plasma ammonia clearance. They also identified that fructose upregulates expression of ChREBP, a transcription factor known as carbohydrate responsive element binding protein, and Slc30a10, a transporter that exports Mn from hepatocytes into bile. Finally, the authors showed that hepatocyte deficiency in ChREBP or Slc30a10 abrogates fructose-induced changes in Mn levels. Overall, their data support a model in which excess fructose consumption leads to increased ChREBP-dependent Slc30a10 expression in the liver, leading to altered liver Mn levels.

The authors present interesting findings and employ multiple approaches to interrogate the link between fructose and manganese. However, we do have suggestions for them on how this manuscript can be improved:

1. Are there changes in Mn levels in organs other than the liver upon fructose feeding? Measuring Mn levels in other organs would be very informative. It could indicate if Mn levels are decreased at the whole body level (suggesting an imbalance between absorption and excretion) or perhaps Mn is just redistributed from liver to other organs. Also, the authors measured blood Mn levels by focusing on plasma. Hemolysis can falsely elevate plasma/serum Mn levels. Were there changes in whole blood?

Response: We understand your concerns, and have measured Mn levels in the whole blood as well as other tissues such as kidney, intestine (supplementary Fig.3), and revised the manuscript accordingly. It turns out that they were not affected by high fructose diet.

2. Were bile Mn levels altered in mice treated with fructose? This would be consistent with the increased Slc30a10 expression observed in fructose-treated mice. Also, did the authors assess Mn absorption from the diet? Could decreased absorption contribute to decreased liver Mn levels? Similarly, could the change in fructose levels in the diets impact the bioavailability of Mn for absorption?

Response: Bile Mn⁺ concentrations were increased after normalization to liver mass (revised Fig.1). We performed Mn absorption experiments, and found there was no significant difference in Mn absorption into portal vein after oral gavage of Mn, which was added in Supplementary Fig.4. Given Mn levels were only affected in the liver by fructose overconsumption, we reason that it is less unlikely for the bioavailability of Mn for absorption is compromised by dietary fructose, which is not completely excluded at present.

3. Slc30a10 is expressed in the intestines and can export Mn from enterocytes into the GI tract. Are there changes in Slc30a10 expression in the intestines in fructose-treated mice?

Response: Intestinal Slc30a10 expression was not affected, as shown in Fig.3e,i

4. The experiments with CCl4 lack a 'no treatment' control. As presented, all mice are treated with CCl4.

Response: We set up PBS control for CCl4 in the experiments, and have included the data in the Figure.

5. For the immunofluorescence presented in Figure 4B, is Slc30a10 expressed in canalicular membranes in hepatocytes? It appears as such, but this could be confirmed using markers such as MDR1.

Response: According to your suggestion, we performed the experiments, and found the overexpressed FLAG-tagged Slc30a10 was mainly localized in canalicular membranes (Figure 4c).

6. AAV-mediated Slc30a10 overexpression resulted in increased transaminase activity. Did the authors use saline-treated controls? Could AAV treatment on its own alter AST and ALT levels? Also, are there changes in ammonia clearance upon overexpression?

Response: We compared plasma transaminase levels in the mice treated with PBS control, low dose and high dose of AAV. The low dose of AAV-Slc30a10 resulted in ~7.5-fold increase in Slc30a10 mRNA levels, which is more physiological relative to high-dose group. Increased ALT and AST were only present in the high dose groups (Supplementary Fig.7e). Addition experiments showed that Slc30a10 overexpression impaired ammonia clearance (Fig.4i).

7. For Figure 5B, a statistically significant difference is noted in liver Mn levels in mice on control vs high fructose diets. The levels appear somewhat similar. Can the authors confirm that these groups differ significantly? Perhaps they could use a split y axis so that it is easier to approach the difference in Mn levels in these groups? The same comment applies to panels C and E.

Response: Thank you for your helpful suggestions. We have redrawn the figures. They are statistically different (revised Fig.5c-f).

8. The figure panel labels for figure 4 do not align with the descriptions in the legend.

Response: We have corrected.

9. The authors noted changes in activity of Mn-dependent enzymes without changes in protein levels. Does the metal status of these proteins not impact the turnover of the proteins? Do Mn levels not impact the expression of these proteins either? Has this been addressed in other papers?

Response: Thank you for raising this interesting question. According to our data, the expression of these proteins was not affected by their metal status. To our knowledge,

there is no report about the regulation of Mn-dependent enzymes expression or turnover by their metal status.

10. A recent paper (cited in the manuscript) has shown that hypoxia-inducible factors also regulate Slc30a10 expression. Does fructose impact HIF-dependent Slc30a10 expression?

Response: According to our data from ChREBP knockout mice, we conclude that the impact of fructose on Mn metabolism is ChREBP-dependent. Western blot showed that fructose does not significantly impact HIFa protein expression in the liver (Supplementary Fig.6c).

REVIEWER COMMENTS

Reviewer #1 (Remarks to the Author):

The authors have addressed most of my previous concern with new data and revised text.

Reviewer #2 (Remarks to the Author):

Dear authors,

Thank you for submitting your revised manuscript and addressing our comments. Overall, we appreciate that you made substantial efforts to address our initial comments. We do have some additional comments based upon your revision.

-Thanks for submitting uncropped Hif blots. However, the protein identified as Hif1a is not the correct size. It should be about 110-120kDa, not 70kDA. We recommend including cobalt-treated HepG2 cells as a positive control. Using nuclear preps can help remove many non-specific bands as well. We do appreciate that Hif blots are not straightforward!

-The absorption experiment where authors collected portal vein blood to measure how much gavaged Mn was absorbed should include a control diet. Also, the methods section needs to be revised to cover the absorption experiment. Finally, the gold standard to measure absorption is to use radioisotopes, not excess non-radioactive manganese. By using excess non-radioactive manganese, the authors are looking at the ability of the mouse to absorb excess manganese, not physiologic levels of manganese. Please comment.

Reviewer #2 (Remarks to the Author):

Dear authors,

Thank you for submitting your revised manuscript and addressing our comments. Overall, we appreciate that you made substantial efforts to address our initial comments. We do have some additional comments based upon your revision.

-Thanks for submitting uncropped Hif blots. However, the protein identified as Hif1a is not the correct size. It should be about 110-120kDa, not 70kDa. We recommend including cobalt-treated HepG2 cells as a positive control. Using nuclear preps can help remove many non-specific bands as well. We do appreciate that Hif blots are not straightforward!

Response: We are grateful for your suggestions. We titrated the HIF1a antibody using human hepatocellular cancer cell line Huh7 exposed to 1% hypoxia condition as a positive control, and successfully detected HIF1a in hypoxia-treated Huh7 cells at 110-120 kDa, but failed to detect in either control or KO liver (Supplementary Fig.6c). Furthermore, we added the mRNA levels of Vegf gene, a target of HIF1, which was comparable between the two groups (Supplementary Fig.6d). These data suggest that HIF1a protein level is very low in the liver under physiological condition and not affected by fructose stimulation. We have made the revision accordingly. We also agree that the HIF data are not straightforward.

-The absorption experiment where authors collected portal vein blood to measure how much gavaged Mn was absorbed should include a control diet. Also, the methods section needs to be revised to cover the absorption experiment. Finally, the gold standard to measure absorption is to use radioisotopes, not excess non-radioactive manganese. By using excess non-radioactive manganese, the authors are looking at the ability of the mouse to absorb excess manganese, not physiologic levels of manganese. Please comment.

Response: We used high-starch diet as a control diet for the high-fructose diet, which is a better control than the chow diet, as described at the beginning of 2nd paragraph in the results. We added the Mn absorption experiment in the methods section. Our results suggest that fructose did not change the intestinal Mn absorption capacity. We agree that radioisotope is more reliable and physiological method. However, it is not available to us at present. Given the circumstance that we did not find significant change of Mn contents beyond the liver in fructose-exposed mice (Supplementary Fig. 3c-f), we did not elaborate

to analyze intestinal Mn absorption using radioisotope. We believe this does not compromise our conclusion about the regulation of liver Mn metabolism by fructose through ChREBP/Slc30a10 pathway.

REVIEWERS' COMMENTS

Reviewer #2 (Remarks to the Author):

Thanks for addressing our comments!